# Multi-Point Collaborative Authentication Method Based on User Image Intelligent Collection in the Internet of Things

**Yunfa Li** [1,*] **, Yifei Tu** [1] **and Jiawa Lu** [2]

[1] Key Laboratory of Complex Systems Modeling and Simulation, School of Computer Science and Technology, Hangzhou Dianzi University, Hangzhou 310018, China

[2] Department of Mechanical, Materials and Manufacturing Engineering, Faculty of Science and Engineering, University of Nottingham Ningbo China, Ningbo 315100, China

[*] Correspondence: yunfali@hdu.edu.cn

**Abstract:** With the increasing demand for intelligent services of the Internet of Things (IoT), its security issues have attracted widespread attention recently. Since most of the existing identity authentication policies are based on a single authentication mode, they are highly likely to cause problems such as illegal operation and stealing of sensor information. In order to meet the needs of increasing IoT users for the security management of intelligent services, a multi-point collaborative authentication method based on user image intelligent collection for the security problems faced by IoT in identity authentication is proposed in the paper. This method firstly collects the identity of the legal user through the intelligent collection technology and then realizes the identity authentication of the unidentified user through the collaborative authentication between the local domain management machine, the back-end image management machine, and the cloud server. Compared with the traditional single identity authentication method, our method uses three-party collaborative authentication to avoid the problem of sensor information stealing easily caused by a single authentication method, which makes the user's identity authentication more secure and effective. The security analysis shows that the method is able to resist multiple attacks and prevent the sensor information from being illegally operated and stolen, protecting the security of the sensor information.

**Keywords:** Internet of Things; identity authentication; collaborative authentication; security

---

## 1. Introduction

With the continuous development of technology, the development of the Internet of Things (IoT) has shown an exponential growth. Combining sensor technology, Internet technology, and wireless technology, IoT realizes real-time interaction between the virtual network and the real world. It senses and collects data in real time through a large number of sensors and transmits data to the server for data calculation and processing. In addition, the processed information is transmitted to the user. IoT has unlimited application prospects and is currently widely used in smart homes, wearable devices, implantable devices, medical devices, connected cars, and transportation systems. Therefore, the Internet of Everything has become an inevitable trend in technology development and industrial application. Although IoT has greatly improved the level of intelligence and automation of society, the information transmitted wirelessly and exposed to the public is highly likely to be tampered with, stolen, and interfered. Therefore, the security of the IoT system has been greatly threatened. Incidents caused by IoT security occur frequently in the real world, and their influence and destructive power are extremely great, so that IoT security has already become a topic of global

concern. The issue of IoT security is the primary problem solved by the development of IoT. With the increasing attention paid to IoT data security, a secure and effective identity authentication protocol has become an important requirement for the rapid development of IoT.

IoT identity authentication is threatened by the following major aspects in terms of security. (1) Denial of service (DoS) attacks: When the data are transmitted, the data transmission of a large number of machines may cause network congestion, because IoT has a large number of nodes and exists in a cluster. The attacker may broadcast invalid information to perform a consumptive attack on the network bandwidth, so that the request of the legal user cannot be executed. (2) Node attack: There is a large number of sensing nodes in the IoT application, most of which are deployed in unattended scenarios. Attackers can easily destroy these nodes and impersonate legitimate nodes. Therefore, there will be a large number of damaged nodes and malicious nodes in the IoT. (3) Replay attack: The attacker can deceive the IoT system to obtain an authenticated identity by sending a packet when the destination host has accepted. (4) Eavesdropping and camouflage attacks: The attacker steals security information from a common channel and falsifies other users' information through known security information. In order to solve these problems and protect the security of identity authentication, this paper proposes a multi-point collaborative authentication method based on user image intelligent collection in the IoT. Firstly, the intelligent collection technology is used to realize the image identity collection of legal users, that is the image information of the legal user is collected by the camera and stored in the database of the local domain management machine, the back-end image management machine, and the cloud server. Secondly, the camera is used to collect the unidentified user image information, which was transmitted to the three databases, as well. Thirdly, the legal user image information and the unidentified user image information in the three databases are compared to realize the identity authentication of the unidentified user, respectively. Not covering the field of imagery, the limitation of our method is that it just compares the image of the legal user with the that of the unidentified user, so there is no algorithm for the image in our method. The security analysis shows that the method is able to resist multiple attacks and prevent the sensor information from being illegally operated and stolen, protecting the security of the sensor information.

## 2. Related Work

With the rapid development of IoT technology, secure identity authentication has become the focus of research for IoT security in the recent years.

Smart card technology and cryptosystems are used in most identity authentication methods currently. Tai et al. [1] analyzed Turkanović's Internet of Things-based authentication and key agreement method for heterogeneous wireless sensor networks (WSN). Analysis showed that the method had two fatal flaws, that is user anonymity was not achieved, and the attacker could easily obtain a session key shared between a normal sensor node and a damaged sensor node that has been connected. In order to solve these two shortcomings, Tai proposed an improved Internet of Things-based authentication and key agreement method. Using smart card technology and a user password system, user and sensor node authentication can be realized with the help of network nodes. Through security analysis, the method has the advantages of user anonymity, ensuring the correctness of the session key and lightweight computing operations. In recent years, more and more identity authentication methods have combined smart card technology with other identification technologies. Challa et al. [2] proposed a new signature-based authentication key establishment method by using Burrows–Abadi–Needham logic for the IoT environment, which can resist various known attacks. The method uses smart card technology, combined with the user's personal biometric technology Bioi and the user's password to achieve mutual authentication between the user and sensor equipment. The method has the advantages of high security, low computing cost, and efficient communication cost; in addition, it is suitable for practical applications in the IoT environment. Hu et al. [3] proposed a security and privacy protection method in the IoT environment, outlined the face recognition and resolution framework based on fog computing, and summarized security and privacy issues, then

proposed an authentication and session key agreement method, a data encryption method, and a data integrity check method in order to solve the confidentiality, integrity, and usability problems in the face recognition process. Dhillon and Kalra [4] used biometrics technology to communicate with VoIP based on the session initiation protocol (SIP) and proposed a new authentication method based on multi-factor ECC. The method uses three users' personal biometrics to provide strong identity checks and thus enhanced security, proving that the method is resistant to a variety of potential threats by conducting rigorous security analysis. Zhang et al. [5] proposed a method for mobile terminal identity authentication based on two-dimensional code technology in a cloud computing environment. The method adopts QR encoding technology as the two-dimensional code processing technology and uses the QR code as the information transmission carrier to realize dynamic authentication of a mobile terminal. According to the security analysis, the method is simple in structure. The method does not require the use of third-party equipment and has high security and adaptability.

The elliptic curve encryption (ECC) algorithm has the advantages of a small key and high computational efficiency. Many researchers use the ECC algorithm to replace the traditional encryption algorithm in the identity authentication method. AL-Turjman et al. [6] proposed a seamless secure authentication and key agreement (S-SAKA) method using bilinear pairing and elliptic curve cryptosystems. The method introduced a mobile receiver strategy to extend user authentication in cloud-based environments. The results showed that the proposed S-SAKA method satisfied the security attributes and was flexible for node-capture attacks. In addition, the method also resisted a large number of well-known potential attacks related to data confidentiality, mutual authentication, session key agreement, user anonymity, password guessing, and key emulation. Kalra and Sood [7] proposed a secure ECC-based mutual authentication protocol for secure communication between embedded devices and cloud servers using Hypertext Transfer Protocol (HTTP) cookies. Through security analysis, the protocol is robust to multiple security attacks and provides basic security requirements. Mo et al. [8] analyzed a secure and efficient user authentication and key agreement protocol (AKAP), and proposed a more efficient remote user mutual AKAP scheme using ECC with a provable security for mobile client–server environments. The proposed scheme not only provides mutual authentication, but also implements a session key agreement between the client and the server. An informal security analysis showed that the scheme protects against well-known attacks and provides anonymity to users.

Multi-factor authentication can enhance the security of the identity authentication process. More and more researchers have improved the traditional single-factor authentication into multi-factor authentication in identity authentication methods. Jiang et al. [9] analyzed a three-factor mutual authentication protocol proposed by Amin for wireless sensor networks, indicating that the protocol was likely to suffer from smart card loss attacks, and the user identity and password could be guessed using brute force techniques. They proposed a lightweight and secure user authentication protocol based on the Rabin cryptosystem. The protocol can resist all possible attacks and provide the required security functions by mutual authentication among users, gateway nodes, and server nodes. The protocol was able to defend against all possible attacks and provide the required security functions. Wazid et al. [10] designed a new secure lightweight three-factor remote user authentication method for the hierarchical IoT network (HIoTN), which is called the user authentication key management protocol (UAKMP). The three factors used in UAKMP are user smart cards, passwords, and personal biometrics. The method provides a variety of features, including offline sensing node registration, freely password, biometric update facility, user anonymity, and perceived node anonymity. Xu and Wu [11] proposed a new three-factor authentication method for wireless sensor networks with a multi-gateway architecture. The method achieved mutual authentication of four factors, which were user, home gateway node, foreign gateway node, and sensor node, respectively. After the formal verification of a protocol security analysis tool (ProVerif) and based on the results of informal analysis, the method was able to defend against various attacks and meet security attributes.

Dynamic identity authentication uses multiple encryption to protect data transmission. More and more researchers have improved the traditional static identity authentication into dynamic identity authentication in identity authentication methods. Gong et al. [12] proposed an IoT-aware node authentication mechanism based on dynamic metrics. Firstly, by introducing the computing functions such as the trust function, the credibility risk assessment function, the feedback control function, and the active function of the sensing node, the dynamic credibility measurement of the sensing node was realized. The dynamic credibility measure of the multi-dimensional sensing node was able to effectively describe the change of the perceived value of the perceived node. Afterwards, a trusted attestation based on the node trusted measure was realized by using the revocable group signature mechanism of a local verifier. Zhang and Xu [13] proposed a secure authentication technology based on a dynamic Bayesian network combined with a trusted protocol in the IoT. By introducing a secure authentication mechanism based on the combined public key and trusted measurement in the network, the security of the information exchange was enhanced. The node credibility and path reliability were considered in the routing decision, so that a highly secure and reliable path was selected for information transfer in the IoT. The evaluation results showed that the proposed algorithm had better security performance than the compared algorithm in terms of the overhead and computational complexity of the real-time application, and it also had the adaptive ability to respond quickly to denial of service attacks and effectively suppress the threat of abnormal entities in the IoT. Xie et al. [14] proposed a new lattice-based dynamic group signature method. This method allows any user to dynamically join a group while achieving effective undo. In addition, the method can achieve non-framework security, ensuring that other users in the system cannot forge the signature of any user. It was proven that the method based on the hardness of the lattice problem in the random prediction model was safe.

As a key part of the development of the Internet of Things, RFID technology has also been the focus of researchers in the field of identity authentication in the Internet of Things in recent years. Shen et al. [15] analyzed the RFID authentication methods of Chen et al. and pointed out that their methods were vulnerable to replay attacks and server spoofing attacks. They proposed a new RFID authentication method using elliptic curve cryptography (ECC). Security analysis showed that the method can meet the security requirements of RFID authentication and does not require additional performance costs, which is more suitable for practical applications. Fan et al. [16] proposed an ultra-lightweight RFID authentication scheme called ULRAS in order to reduce the computational cost. ULRAS uses only bit and XOR operations to prevent DDOS attacks. ULRAS uses subkeys and sub-indexes throughout the key update process and uses the RR method throughout the protocol to make the protocol well resistant to various attacks. For each update process, ULRAS only needs to update a portion of the key $K$ randomly, which makes the update process more random. Compared to other protocols, ULRAS reduces the cost of computing and communication resources and is more secure. Aghili et al. [17] evaluated the security of the ultra-lightweight RFID mutual authentication protocol (ULRMAPC) proposed by Fan and demonstrated that the protocol was vulnerable to denial of service, reader and tag emulation, and desynchronization attacks. They proposed a new improved authentication, which can provide sufficient resistance to known active and passive attacks.

In summary, most of the research on IoT authentication in the recent years still uses the smart card technology identity or the single identity authentication method. The identity authentication method using the smart card technology stores the user's private information in the smart card chip, and the attacker can easily steal the user information in the smart card, while the smart card is easily lost or damaged. The single identity authentication method does not achieve multi-end mutual authentication and is vulnerable to server spoofing attacks and spoofing attacks. A multi-point collaborative authentication method based on user image intelligent collection in the IoT is proposed in this paper. Compared with the traditional single identity authentication method, our method uses three-party collaborative authentication, which is able to avoid the problems of sensor information theft and server spoofing attacks, frequently occurring in the single authentication method. Compared with the traditional identity authentication method using smart card technology, the method proposed

in this paper using image intelligent collection technology is more friendly and direct and can avoid the problem of smart card information loss. Although the method does not use the password system, the user image is used as the unique identity information. Attacks on user password information, such as password guessing, are avoided.

The organization of this paper is as follows: The specific method of the multi-point collaborative authentication method is introduced in the Section 3. The safety and cost of the method are analyzed in the Section 4. The Section 5 presents conclusions and future work.

## 3. Multi-Point Collaborative Authentication

This section proposes a multi-point collaborative authentication method based on user image intelligent collection in the IoT. The method mainly includes four phases: (1) the system establishment phase of user image intelligent multi-point collaborative authentication; (2) the authentication phase of the local domain management machine and the back-end image management machine; (3) the authentication phase of the local domain management machine and the cloud server; (4) the authentication phase of the back-end image management machine and the cloud server. Through these four phases, it is possible to realize the security of user image intelligent collection and its service intelligent control.

In these phases, the information transmission between the local domain management machine, the back-end image management machine, and the cloud server follows the SSL (secure socket layer) or TLS (transport later security) protocol. Its architecture diagram is shown in Figure 1. The symbols and definitions used in this paper are shown in Table 1.

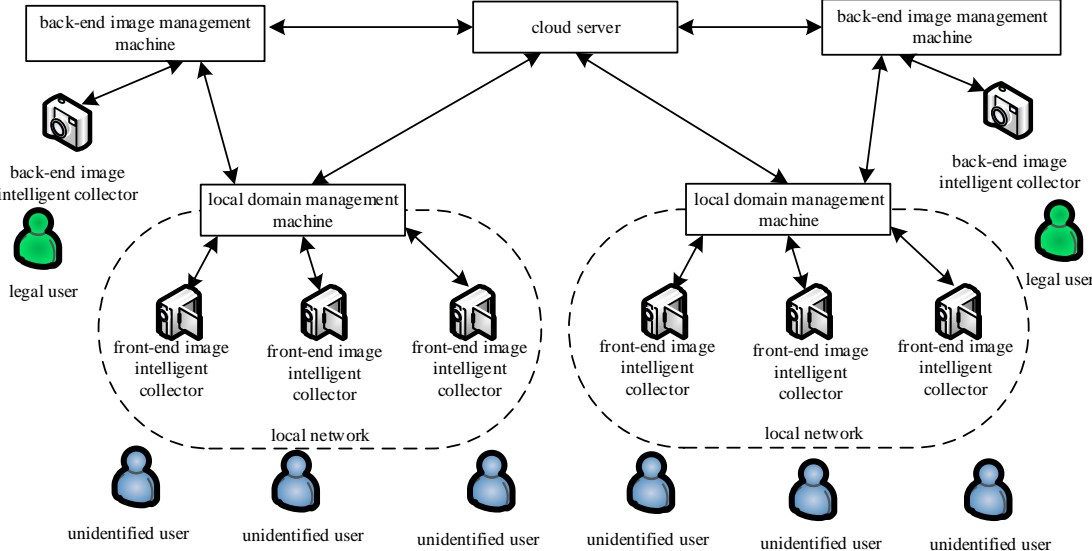

**Figure 1.** Architecture diagram of the multi-point collaborative authentication method based on user image intelligent collection in the IoT.

**Table 1.** Symbols and definitions.

| Symbol | Definition |
| --- | --- |
| $LDMM_j$ | The $j^{\text{th}}$ local domain management machine |
| $BIMM_j$ | The $j^{\text{th}}$ back-end image management machine |
| $BIIC_j$ | The $j^{\text{th}}$ back-end image intelligent collector |
| $CS$ | Cloud server |
| $FIIC_k$ | The $k^{\text{th}}$ front-end image intelligent collector |
| $U$ | Legal user |
| $U^*$ | Unidentified user |
| $U_i$ | The $i^{\text{th}}$ legal user |
| $T_i$ | The $i^{\text{th}}$ timestamp value generated by the front-end image intelligent collector |
| $TS_i$ | The $i^{\text{th}}$ timestamp value generated by the local domain management machine |
| $ID(A)$ | $A$'s identity information |
| $PK_k(A)$ | The $k^{\text{th}}$ public key generated by $A$ |
| $E(A)_{PK_k(B)}$ | Encrypt $A$ with the $k^{\text{th}}$ public key generated by $B$ |
| $SK_k(A)$ | The $k^{\text{th}}$ private key generated by $A$ |
| $D(A)_{SK_k(B)}$ | Decrypt $A$ with the $k^{\text{th}}$ private key generated by $B$ |
| $S(A)_{SK_k(B)}$ | Sign $A$ with the $k^{\text{th}}$ private key generated by $B$ |
| $V(A)_{PK_k(B)}$ | Verify $A$ with the $k^{\text{th}}$ public key generated by $B$ |
| $P(U_i)$ | Image of legal user $U_i$ |
| $A \models P(B)$ | $A$ captures $B$ image |
| $(A \updownarrow B) \to C$ | $A$ transfers $B$ to $C$ |
| $C \leftarrow (A \updownarrow B)$ | $C$ receives $A$ transmitted by $B$ |
| $DB(A)$ | $A$'s database |
| $A \Uparrow DB(B)$ | $A$ build a database containing $B$ information |
| $A||B$ | The parallel operation of $A$ and $B$ |
| $\Phi(B)$ | Demand $B$ |
| $(A \updownarrow \Phi(B)) \to C$ | $A$ transmits demand $B$ to $C$ |
| $ACK$ | Confirmation message |
| $A \triangleright \triangleleft B$ | $A$ check $B$ |
| $A \otimes B$ | $A$ produces $B$ |
| $A \overset{?}{=} B$ | Whether $A$ is equal to $B$ |
| $A \oplus B$ | $A$ stores $B$ |

*3.1. The System Establishment Phase of User Image Intelligent Multi-Point Collaborative Authentication*

At this phase, the back-end image intelligent collector intelligently collects the image of the legal user and transmits it to the back-end image management machine. Then, the back-end image management machine requests the back-end manager to input the identity information and uses a secure encryption algorithm to transmit it to the local domain management machine and the cloud server through the secure channel. Finally, they establish their own image information database in the back-end image management machine, local domain management machine, and cloud server to complete the system establishment. The brief process diagram is shown in Figure 2. The algorithm is shown in Algorithm 1.

---

**Algorithm 1:** The System Establishment Phase of User Image Intelligent Multi-Point Collaborative Authentication

---

1. $BIIC_j | \Rightarrow P(U_i), (BIIC_j \updownarrow P(U_i)) \rightarrow BIMM_j$.
2. $BIMM_j \leftarrow (BIIC_j \updownarrow P(U_i)), BIMM_j | \Rightarrow ID(BIMM_j), BIMM_j | \Rightarrow ID(U_i)$.
3. $BIMM_j \Uparrow DB(U_i||P(U_i)||ID(U_i)||BIMM_j||ID(BIMM_j))$.
4. $PK_1(BIMM_j), SK_1(BIMM_j), E(P(U_i)||ID(BIMM_j))_{PK_1(BIMM_j)}$,
   $(BIMM_j \updownarrow E(P(U_i)||ID(BIMM_j))_{PK_1(BIMM_j)}) \rightarrow LDMM_j$.
5. $LDMM_j \leftarrow (BIMM_j \updownarrow E(P(U_i)||ID(BIMM_j))_{PK_1(BIMM_j)}), (LDMM_j \updownarrow$
   $\Phi(SK_1(BIMM_j))) \rightarrow BIMM_j$.
6. $BIMM_j \leftarrow (LDMM_j \updownarrow \Phi(SK_1(BIMM_j))), (BIMM_j \updownarrow SK_1(BIMM_j)) \rightarrow LDMM_j$.
7. $LDMM_j \leftarrow (BIMM_j \updownarrow SK_1(BIMM_j)), D(E(P(U_i)||ID(BIMM_j))_{PK_1(BIMM_j)})_{SK_1(BIMM_j)}$.
8. $LDMM_j \Uparrow DB(P(U_i)||BIMM_j||ID(BIMM_j))$.
9. $LDMM_j \triangleright \triangleleft ((LDMM_j \Uparrow DB(P(U_i)||BIMM_j||ID(BIMM_j))) \overset{?}{=} 1)$, if it is not one, then go to
   (8), else $(LDMM_j \updownarrow ACK) \rightarrow BIMM_j$.
10. $BIMM_j \leftarrow (LDMM_j \updownarrow ACK), PK_2(BIMM_j), SK_2(BIMM_j)$.
11. $E(P(U_i)||ID(U_i)||ID(BIMM_j))_{PK_2(BIMM_j)}, (BIMM_j \updownarrow$
    $E(P(U_i)||ID(U_i)||ID(BIMM_j))_{PK_2(BIMM_j)}) \rightarrow CS$.
12. $CS \leftarrow (BIMM_j \updownarrow (E(P(U_i)||ID(U_i)||ID(BIMM_j))_{PK_2(BIMM_j)}), (CS \updownarrow$
    $\Phi(SK_2(BIMM_j))) \rightarrow BIMM_j$.
13. $BIMM_j \leftarrow (CS \updownarrow \Phi(SK_2(BIMM_j))), (BIMM_j \updownarrow SK_2(BIMM_j)) \rightarrow CS$.
14. $CS \leftarrow (BIMM_j \updownarrow SK_2(BIMM_j)), D(P(U_i)||ID(U_i)||ID(BIMM_j))_{PK_2(BIMM_j)})_{SK_2(BIMM_j)}$.
15. $CS \Uparrow DB(P(U_i)||ID(U_i)||ID(BIMM_j))$.
16. $CS \triangleright \triangleleft (DB(P(U_i)||ID(U_i)||ID(BIMM_j)) \overset{?}{=} 1)$, if it is not one, then go to (15), else
    $(CS \updownarrow ACK) \rightarrow BIMM_j$.
17. $BIMM_j \leftarrow (CS \updownarrow ACK), BIMM_j \triangleright \triangleleft (\exists (U_i||P(U_i)||ID(U_i)) \overset{?}{=} 1)$, if it is one, then go to (1),
    else go to (18).
18. End.

---

The specific process execution is described as follows:

Step 1: The back-end image intelligent collector $BIIC_j$ ($j$ = 1, 2, 3, ..., n) intelligently collects the image $P(U_i)$ of the legal user $U_i$ ($i$ = 1, 2, 3, ..., n) according to the requirements of the back-end manager. The legal user image $P(U_i)$ is transmitted to the corresponding back-end image management machine $BIMM_j$ via the secret network.

Step 2: After receiving the legal user image $P(U_i)$ transmitted by the back-end image intelligent collector, the back-end image management machine $BIMM_j$ requests the back-end manager to input the identity information $ID(BIMM_j)$ of the back-end image management machine $BIMM_j$ and the identity information $ID(U_i)$ of this legal user.

Step 3: After receiving the identity information $ID(U_i)$ of the input legal user, the back-end image management machine $BIMM_j$ constructs a corresponding back-end image management information database $DB(U_i||P(U_i)||ID(U_i)||BIMM_j||ID(BIMM_j))$. This database contains the identity information $ID(BIMM_j)$ of the back-end image management machine, the legal user name $U_i$, the identity information $ID(U_i)$ of the legal user, and the image information $P(U_i)$ of the legal user.

Step 4: After constructing the database $DB(U_i||P(U_i)||ID(U_i)||BIMM_j||ID(BIMM_j))$, the back-end image management machine $BIMM_j$ first generates a public key $PK_1(BIMM_j)$ and a corresponding private key $SK_1(BIMM_j)$ based on the elliptic curve encryption method. On this basis, the back-end image management machine $BIMM_j$ encrypts the image $P(U_i)$ of the legal user and its own identity information $ID(BIMM_j)$ using the public key $PK_1(BIMM_j)$ based on the elliptic

curve encryption method. The encrypted file is sent to the local domain management machine via SSL or TLS.

Step 5: After receiving the encrypted file $(P(U_i)||ID(BIMM_j))_{PK_1(BIMM_j)}$, the local domain management machine applies it to the back-end image management machine $BIMM_j$ for the private key $SK_1(BIMM_j)$.

Step 6: After receiving the private key application of the local domain management machine, the back-end image management machine $BIMM_j$ sends the private key $SK_1(BIMM_j)$ to the local domain management machine via SSL or TLS.

Step 7: After receiving the private key $SK_1(BIMM_j)$ sent by the back-end image management machine $BIMM_j$, the local domain management machine decrypts the received encrypted file $(P(U_i)||ID(BIMM_j))_{PK_1(BIMM_j)}$.

Step 8: After decrypting the encrypted file $(P(U_i)||ID(BIMM_j))_{PK_1(BIMM_j)}$, the local domain management machine construct a corresponding local domain management image information database $DB(P(U_i)|BIMM_j||ID(BIMM_j))$. This database contains image information $P(U_i)$ of the legal user, the back-end image management machine name $BIMM_j$, and the identity information $ID(BIMM_j)$ of the back-end image management machine $BIMM_j$.

Step 9: The local domain management machine determines whether the local domain management image information database $DB(P(U_i)||BIMM_j||ID(BIMM_j))$ is constructed. If not, go the Step 8; otherwise, send confirmation message to the back-end image management machine $BIMM_j$.

Step 10: After receiving the confirmation message sent by the local domain management machine, the back-end image management machine $BIMM_j$ generates a public key $PK_2(BIMM_j)$ and a corresponding private key $SK_2(BIMM_j)$ according to the elliptic curve encryption method.

Step 11: The back-end image management machine $BIMM_j$ uses the public key $PK_2(BIMM_j)$ to encrypt image $P(U_i)$ of the legal user, the identity information $ID(U_i)$ of the user image, and its own identity information $ID(BIMM_j)$ according to the elliptic curve encryption method, then sends its encrypted file $(P(U_i)||ID(U_i)||ID(BIMM_j))_{PK_2(BIMM_j)}$ to the cloud server via SSL or TLS.

Step 12: After receiving the encrypted file $(P(U_i)||ID(U_i)||ID(BIMM_j))_{PK_2(BIMM_j)}$, the cloud server applies it to the back-end image management machine $BIMM_j$ for the private key $SK_2(BIMM_j)$.

Step 13: After receiving the private key application of the cloud server, the back-end image management machine $BIMM_j$ sends the private key $SK_2(BIMM_j)$ to the cloud server via SSL or TLS.

Step 14: After receiving the private key $SK_2(BIMM_j)$ sent by the back-end image management machine $BIMM_j$, the cloud server decrypts the received encrypted file $(P(U_i)||ID(U_i)||ID(BIMM_j))_{PK_2(BIMM_j)}$.

Step 15: After decrypting the encrypted file $(P(U_i)||ID(U_i)||ID(BIMM_j))_{PK_2(BIMM_j)}$, the cloud server constructs a corresponding cloud server image information database $DB(P(U_i)||ID(U_i)||ID(BIMM_j))$. This database contains the image information $P(U_i)$ of the legal user, the identity information $ID(U_i)$ of the legal user, and the identity information $ID(BIMM_j)$ of the back-end image management machine $BIMM_j$.

Step 16: The cloud server determines whether the cloud server image information database $DB(P(U_i)||ID(U_i)||ID(BIMM_j))$ is constructed. If not, go to Step 15; otherwise, send a confirmation message to the back-end image management machine $BIMM_j$.

Step 17: After receiving the confirmation message sent by the cloud server, the back-end image management machine $BIMM_j$ determines whether the back-end manager needs the back-end image intelligent collector $BIIC_j$ to collect the legal user image, and if necessary, then go to the Step 1; otherwise, go to Step 18.

Step 18: End of system construction.

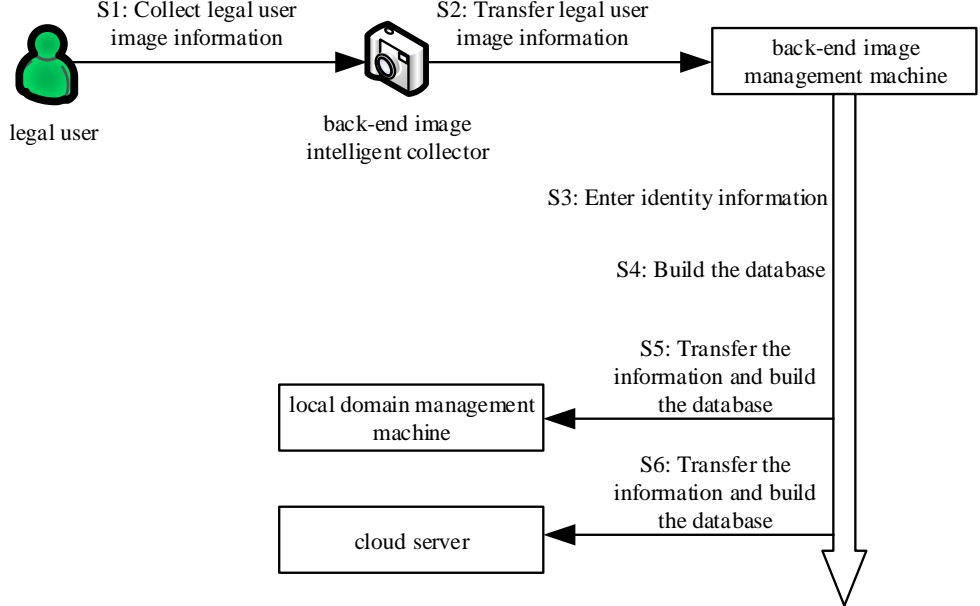

**Figure 2.** The system establishment phase's brief process diagram.

*3.2. The Authentication Phase of the Local Domain Management Machine and the Back-End Image Management Machine*

At this phase, the front-end image intelligent collector collects the unidentified user image information and transmits it to the local domain management machine via the secret network. The local domain management machine applies for authentication to the back-end image management machine in the form of a digital signature. Then, the local domain management machine encrypts the unidentified user image information and transmits it to the back-end image management machine through the encryption algorithm. Next, query the image information of all legal users in the local domain management image information database and the back-end image management image information database and respectively compare the legal user image information in these two databases with the image information of the unidentified user. Finally, complete mutual authentication between the local domain management machine and the back-end image management machine. The brief process diagram is shown in Figure 3. The algorithm is shown in Algorithm 2.

---

**Algorithm 2:** The Authentication Phase of the Local Domain Management Machine and the Back-End Image Management Machine

---

1. $FIIC_k \triangleright \triangleleft (\exists (U^*||P(U^*)) \overset{?}{=} 1)$, if it is one, then go to (3), else go to (2).

2. $FIIC_k$ wait 3 s, then go to (1).

3. $FIIC_k| \Rightarrow P(U_n^*), FIIC_k \otimes T_i, (FIIC_k \updownarrow (P(U_n^*)||T_i) \rightarrow LDMM_j$.

4. $LDMM_j \leftarrow (FIIC_k \updownarrow (P(U_n^*)||T_i), LDMM_j \otimes TS_i, LDMM_j \triangleright \triangleleft (((T_i - TS_i) \geq \Delta t) \overset{?}{=} 1)$, if it is one, $LDMM_j$ delete $P(U_n^*)$, then go to (2), else

   $LDMM_j \triangleright \triangleleft (\exists P(U_i) \in (DB(P(U_i)||BIMM_j||ID(BIMM_j))), LDMM_j \triangleright \triangleleft (P(U_i) \overset{?}{=} P(U_n^*))$, if $\exists P(U_i) \in (DB(P(U_i)||BIMM_j||ID(BIMM_j)))$ and $P(U_i) = P(U_n^*)$, then go to (5), else delete $P(U_n^*)$, and go to (2).

5. $message1 = (\text{Apply for authentication}), S(message1)_{SK(LDMM_j)},$
   $(LDMM_j \updownarrow S(message1)_{SK(LDMM_j)}) \rightarrow BIMM_j$.

6. $BIMM_j \leftarrow (LDMM_j \updownarrow S(message1)_{SK(LDMM_j)}), V(S(message1)_{SK(LDMM_j)})_{PK(LDMM_j)} \overset{?}{=} 1$, if it is one, then go to (7), else show "Apply for authentication failure", and go to (3).

7. $message =$
   (The authentication was successful, please transmit the image of the unidentified user),
   $(BIMM_j \updownarrow E(message)_{PK_1(BIMM_j)}) \rightarrow LDMM_j$.

8. $LDMM_j \leftarrow (BIMM_j \updownarrow E(message)_{PK_1(BIMM_j)}), D(E(message)_{PK_1(BIMM_j)})_{SK_1(BIMM_j)}$.

9. $S(P(U_n^*))_{SK(LDMM_j)}, (LDMM_j \updownarrow S(P(U_n^*))_{SK(LDMM_j)}) \rightarrow BIMM_j$.

10. $BIMM_j \leftarrow (LDMM_j \updownarrow S(P(U_n^*))_{SK(LDMM_j)}), V(S(P(U_n^*))_{SK(LDMM_j)})_{PK(LDMM_j)} \overset{?}{=} 1$, if it is one, then go to (11), else show "The verification of the image of the unidentified user failed", and go to (3).

11. $BIMM_j \leftarrow (FIIC_k \updownarrow P(U_n^*))$.

12. $BIMM_j \triangleright \triangleleft (\exists P(U_i) \in DB(U_i||P(U_i)||ID(U_i)||BIMM_j||ID(BIMM_j))$,
    $BIMM_j \triangleright \triangleleft (P(U_i) \overset{?}{=} P(U_n^*))$, if $\exists P(U_i) \in (DB(U_i||P(U_i)||ID(U_i)||BIMM_j||ID(BIMM_j)))$ and $P(U_i) = P(U_n^*)$, then show "The verification is success between the local domain management machine and the back-end image management machine", and go to (13), else show "The verification failed between the local domain management machine and the back-end image management machine", and go to (15).

13. message = (The verification was successful between the local domain management machine and the back-end image management machine),
    $(BIMM_j \updownarrow E(message)_{PK(BIMM_j)}) \rightarrow LDMM_j$.

14. $BIMM_j \triangleright \triangleleft$ the transmission of the message is over. If it is over, then go to (15), else go to (13).

15. End.

---

The specific process execution is described as follows:

Step 1: The front-end image intelligent collector $FIIC_k$ intelligently judges whether there is an unidentified user who needs image collection according to the surrounding scenes. If needed, then go to Step 3, otherwise, go to Step 2.

Step 2: The front-end image intelligent collector $FIIC_k$ waits for three seconds and then goes to Step 1.

Step 3: The front-end image intelligent collector $FIIC_k$ intelligently collects the image $P(U_n^*)$ of the unidentified user according to the surrounding scene and generates a timestamp value $T_i$. Then, the image of the unidentified user and the current timestamp value $(P(U_n^*)||T_i)$ are sent to the local domain management machine through the secret network.

Step 4: After receiving the image of the unidentified user and the current timestamp value $(P(U_n^*)||T_i)$ sent by the front-end image intelligent collector $FIIC_k$, the local domain management machine generates a timestamp value $TS_i$. Firstly, check if the session delay $T_i - TS_i$ is within the allowable time interval $\Delta t$. If $(T_i - TS_i) \geq \Delta t$, the session times out, and delete the image $P(U_n^*)$ of the unidentified user sent by the front-end image intelligent collector $FIIC_k$, then go to Step 2. Then, query the image information $P(U_i)$ of all legal users in the local domain management image information database $DB(P(U_i)||BIMM_j||ID(BIMM_j))$, and compare the legal user image information $P(U_i)$ in the database $DB(P(U_i)||BIMM_j||ID(BIMM_j))$ with the image $P(U_n^*)$ of the unidentified user sent by the front-end image intelligent collector $FIIC_k$. If the image information $P(U_i)$ of a certain legal user exists in the local domain management image information database $DB(P(U_i)||BIMM_j||ID(BIMM_j))$ and the image $P(U_n^*)$ of the unidentified user sent by the front-end image intelligent collector $FIIC_k$ is the same (i.e., $P(U_n^*) = P(U_i)$), go to Step 5; otherwise, delete the image $P(U_n^*)$ of the unidentified user sent by the front-end image intelligent collector $FIIC_k$, then go to Step 2.

Step 5: According to the back-end image management machine $BIMM_j$ corresponding to the image information $P(U_i)$ of the legal user in the database $DB(P(U_i)||BIMM_j||ID(BIMM_j))$, the local domain management machine first uses the private key $SK(LDMM_j)$ and signs the "Apply for authentication" message, i.e., ("Apply for authentication")$_{SK(LDMM_j)}$. Then, it sends the signed message to the back-end image management machine $BIMM_j$.

Step 6: After receiving the signature message ("Apply for authentication")$_{SK(LDMM_j)}$ sent by the local domain management machine, the back-end image management machine $BIMM_j$ authenticates the signature message by using the public key of the local domain management machine. If the authentication is successful, go to Step 7. Otherwise, display "Apply for authentication failure", and go to Step 3.

Step 7: The back-end image management machine $BIMM_j$ encrypts the "The authentication is successful, please transmit the image of the unidentified user" message using the public key $PK_1(BIMM_j)$, according to the elliptic curve encryption method, and then sends the encrypted message ("The authentication is successful, please transmit the image of the unidentified user")$_{PK_1(BIMM_j)}$ to the local domain management machine via SSL or TLS.

Step 8: After receiving the encrypted message ("The authentication is successful, please transmit the image of the unidentified user")$_{PK_1(BIMM_j)}$, the local domain management machine uses private key $SK_1(BIMM_j)$ to decrypt.

Step 9: According to the decrypted message, the local domain management machine first uses the private key $SK(LDMM_j)$ to sign the image $P(U_n^*)$ message of the unidentified user collected by the front-end image intelligent collector $FIIC_k$, i.e., $(P(U_n^*))_{SK(LDMM_j)}$. Then, the signed message is sent to the back-end image management machine $BIMM_j$.

Step 10: After receiving the signature message $(P(U_n^*))_{SK(LDMM_j)}$ sent by the local domain management machine, the back-end image management machine $BIMM_j$ authenticates the signature message by using the public key of the local domain management machine. If the authentication is successful, go to Step 11. Otherwise, display "The verification of the image of the unidentified user failed", and go to Step 3.

Step 11: The back-end image management machine $BIMM_j$ receives the image $P(U_n^*)$ of the unidentified user sent by the front-end image intelligent collector $FIIC_k$.

Step 12: The back-end image management machine $BIMM_j$ first queries the image information $P(U_i)$ of all legal users in its database $DB(U_i||P(U_i)||ID(U_i)||BIMM_j||ID(BIMM_j))$ and compares the legal user image information $P(U_i)$ in the database $DB(U_i||P(U_i)||ID(U_i)||BIMM_j||ID(BIMM_j))$ with the image $P(U_n^*)$ of the unidentified user sent by the front-end image intelligent collector $FIIC_k$. If the image information $P(U_i)$ of a certain legal user exists in the back-end image management information database $DB(U_i||P(U_i)||ID(U_i)||BIMM_j||ID(BIMM_j))$ and the image $P(U_n^*)$ of the unidentified user sent by the front-end image intelligent collector $FIIC_k$ are the same (i.e., $P(U_n^*) = P(U_i)$), the back-end image management machine $BIMM_j$ displays "The verification is successful

between the local domain management machine and the back-end image management machine".
Otherwise, the back-end image management machine $BIMM_j$ displays "The verification failed
between the local domain management machine and the back-end image management machine"
and goes to Step 15.

Step 13: The back-end image management machine $BIMM_j$ encrypts the image authentication
result message "The verification is successful between the local domain management machine and
the back-end image management machine" and then sends its encrypted message to the local domain
management machine via SSL or TLS.

Step 14: The back-end image management machine $BIMM_j$ determines whether the image
authentication result message is sent. If sent, go to Step 15; otherwise, go to Step 13.

Step 15: End.

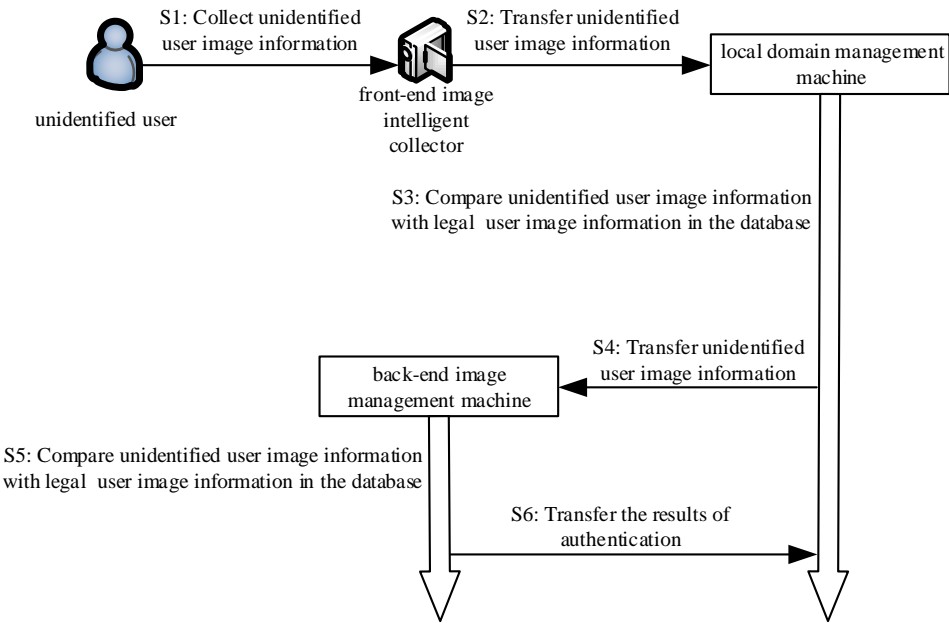

**Figure 3.** The authentication phase of the local domain management machine and the back-end image
management machine brief process diagram.

### 3.3. The Authentication Phase of the Local Domain Management Machine and Cloud Server

In this phase, the local domain management machine applies for authentication to the cloud
server in the form of a digital signature. Then, the local domain management machine encrypts
the unidentified user image information and transmits it to the cloud server through the encryption
algorithm. Then, query the image information of all legal users in the cloud server image information
database, and compare the legal user image information in the database with the image information of
the unidentified user. Finally, complete mutual authentication between the local domain management
machine and the cloud server. The brief process diagram is shown in Figure 4. The algorithm is shown
in Algorithm 3.

---

**Algorithm 3:** The Authentication Phase of the Local Domain Management Machine and Cloud Server

---

1. $LDMM_j \leftarrow (BIMM_j \updownarrow E(message)_{PK(BIMM_j)}), D(E(message)_{PK(BIMM_j)})_{SK(BIMM_j)}.$

2. $LDMM_j \rhd \lhd$The verification of the message is successful, if the verification is successful, then go to (3), else go to (13).

3. $message1 = $ (Apply for authentication), $S(message1)_{SK(LDMM_j)}$, $(LDMM_j \updownarrow S(message1)_{SK(LDMM_j)}) \rightarrow CS.$

4. $CS \leftarrow (LDMM_j \updownarrow S(message1)_{SK(LDMM_j)}), V(S(message1)_{SK(LDMM_j)})_{PK(LDMM_j)} \overset{?}{=} 1,$ if it is one, then go to (5), else show "Apply for authentication failure" and go to (13).

5. $message2 = $ (The authentication is successful, please transmit the image of the unidentified user), $(CS \updownarrow E(message2)_{PK_1(BIMM_j)}) \rightarrow LDMM_j.$

6. $LDMM_j \leftarrow (CS \updownarrow E(message2)_{PK_1(BIMM_j)}), D(E(message2)_{PK_1(BIMM_j)})_{SK_1(BIMM_j)}.$

7. $S(P(U_n^*))_{SK(LDMM_j)}, (LDMM_j \updownarrow S(P(U_n^*))_{SK(LDMM_j)}) \rightarrow CS.$

8. $CS \leftarrow (LDMM_j \updownarrow S(P(U_n^*))_{SK(LDMM_j)}), V(S(P(U_n^*))_{SK(LDMM_j)})_{PK(LDMM_j)} \overset{?}{=} 1.$ If it is one, then go to (9), else show "The authentication of the image of the unidentified user failed", and go to (13).

9. $CS \leftarrow (FIIC_k \updownarrow P(U_n^*)).$

10. $CS \rhd \lhd (\exists P(U_i) \in DB(P(U_i)||ID(U_i)||ID(BIMM_j)), CS \rhd \lhd (P(U_i) \overset{?}{=} P(U_n^*)),$ if $\exists P(U_i) \in (DB(P(U_i)||ID(U_i)||ID(BIMM_j)))$ and $P(U_i) = P(U_n^*)$, then the cloud server shows "The verification is successful between the local domain management machine and the cloud server", and go to (11), else the cloud server shows "The verification failed between the local domain management machine and the cloud server", and go to (13).

11. $message3 = $ (The verification is successful between the local domain management machine and the cloud server), $(CS \updownarrow E(message3)_{PK(BIMM_j)}) \rightarrow LDMM_j$, $(CS \updownarrow E(message3)_{PK(BIMM_j)}) \rightarrow BIMM_j.$

12. $CS \rhd \lhd$the transmission of the message is over. If it is over, then go to (13), else go to (11).

13. End.

---

The specific process execution is described as follows:

Step 1: The local domain management machine receives the image authentication result message sent by the back-end image management machine $BIMM_j$ and decrypts the message.

Step 2: The local domain management machine determines the decrypted image authentication result message. If the authentication with the local domain management machine is successful, the process goes to Step 3. Otherwise, the process goes to Step 13.

Step 3: The local domain management machine first signs the "Apply for authentication" message using the private key $SK(LDMM_j)$, i.e. ("Apply for authentication")$_{SK(LDMM_j)}$, then sends the signed message ("Apply for authentication")$_{SK(LDMM_j)}$ to the cloud server.

Step 4: After receiving the signature message ("Apply for authentication")$_{SK(LDMM_j)}$ sent by the local domain management machine, the cloud server authenticates the signature message ("Apply for authentication")$_{SK(LDMM_j)}$ by using the public key of the local domain management machine. If the authentication is successful, go to Step 5. Otherwise, display "Apply for authentication failure", and go to Step 13.

Step 5: According to the elliptic curve encryption method, the cloud server encrypts the "The authentication is successful, please transmit the image of the unidentified user" message using the public key $PK_1(BIMM_j)$ and then sends the encrypted message ("The authentication is successful, please transmit the image of the unidentified user")$_{PK_1(BIMM_j)}$ to the local domain management machine via SSL or TLS.

Step 6: After receiving the encrypted message ("The authentication is successful, please transmit the image of the unidentified user")$_{PK_1(BIMM_j)}$, the local domain management machine uses the private key $SK_1(BIMM_j)$ to decrypt the message.

Step 7: According to the decrypted message, the local domain management machine first uses the private key $SK(LDMM_j)$ to sign the collected image $P(U_n^*)$ of the unidentified user, i.e., $P(U_n^*)_{SK(LDMM_j)}$, then sends the signature message $(U_n^*))_{SK(LDMM_j)}$ to the cloud server.

Step 8: After receiving the signature message $P(U_n^*))_{SK(LDMM_j)}$ sent by the local domain management machine, the cloud server authenticates the signature message $P(U_n^*))_{SK(LDMM_j)}$ by using the public key of the local domain management machine. If the authentication is successful, go the Step 9. Otherwise, display "The authentication of the image of the unidentified user failed" and go to Step 13.

Step 9: The cloud server receives the image $P(U_n^*)$ of the unidentified user sent by the front-end image intelligent collector $FIIC_k$.

Step 10: The cloud server first queries the image information $P(U_i)$ of all legal users in its database $DB(P(U_i)||ID(U_i)||ID(BIMM_j))$ and compares the legal user image information $P(U_i)$ in the database $DB(P(U_i)||ID(U_i)||ID(BIMM_j))$ with the image $P(U_n^*)$ of the unidentified user sent by the front-end image intelligent collector $FIIC_k$. If the image information $P(U_i)$ of a certain legal user exists in the cloud server database $DB(P(U_i)||ID(U_i)||ID(BIMM_j))$ and the image $P(U_n^*)$ of the unidentified user sent by the front-end image intelligent collector $FIIC_k$ are the same (i.e., $P(U_n^*) = P(U_i)$), the cloud server displays "The verification is successful between the local domain management machine and the cloud server". Otherwise, the cloud server displays "The verification failed between the local domain management machine and the cloud server" and goes to Step 13.

Step 11: The cloud server encrypts the image authentication result message "The verification is successful between the local domain management machine and the cloud server" and then sends the encrypted message to the local domain management machine and the corresponding back-end image management machine $BIMM_j$ via SSL or TLS.

Step 12: The cloud server determines whether the image authentication result message is sent. If sent, go to Step 13; otherwise, go to Step 11.

Step 13: End.

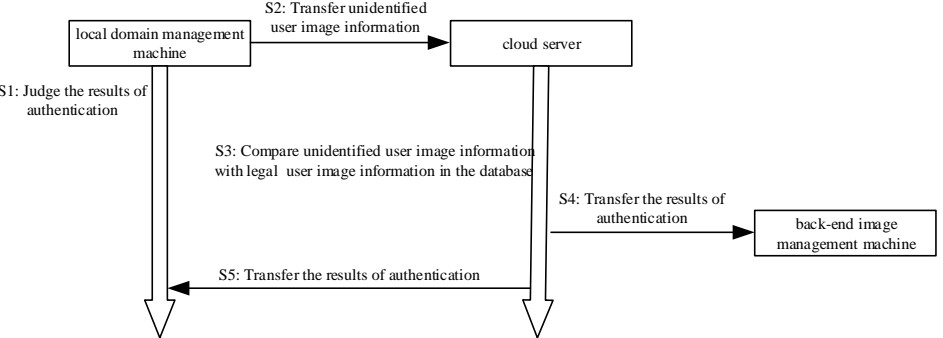

**Figure 4.** The authentication phase of the local domain management machine and cloud server brief process diagram.

### 3.4. The Authentication Phase of the Back-End Image Management Machine and Cloud Server

In this phase, the back-end image management machine applies for authentication to the cloud server in the form of a digital signature. Secondly, the back-end image management machine encrypts the information such as the unidentified user image information and transmits it to the cloud server through the encryption algorithm. Then, query the image information of all legal users in the cloud

server image information database. Next, compare the image information and the identity information with the image of the unidentified user, which has been authenticated by the signature and the identity information of the unidentified user. Finally, complete mutual authentication between the back-end image management machine and the cloud server. The brief process diagram is shown in Figure 5. The algorithm is shown in Algorithm 4.

---

**Algorithm 4:** The Authentication Phase of the Back-End Image Management Machine and Cloud Server

1. $BIMM_j \leftarrow (CS \updownarrow E(message3)_{PK(BIMM_j)}), D(E(message3)_{PK(BIMM_j)})_{SK(BIMM_j)}$.
2. $BIMM_j \triangleright \triangleleft$The verification of the message is success . If the verification is success, then go to (3), else go to (14).
3. $message1$ = (Apply for authentication), $S(message1)_{SK(BIMM_j)}$,
   $(BIMM_j \updownarrow S(message1)_{SK(BIMM_j)}) \rightarrow CS$.
4. $CS \leftarrow (BIMM_j \updownarrow S(message1)_{SK(BIMM_j)}), V(S(message1)_{SK(BIMM_j)})_{PK(BIMM_j)} \overset{?}{=} 1$, if it is one, then go to (5), else show "Apply for authentication failure", and go to (13).
5. $message2$ = (The authentication is successful, please transmit the image of the unidentified user), $(CS \updownarrow E(message2)_{PK_1(BIMM_j)}) \rightarrow BIMM_j$.
6. $BIMM_j \leftarrow (CS \updownarrow E(message2)_{PK_1(BIMM_j)}), D(E(message2)_{PK_1(BIMM_j)})_{SK_1(BIMM_j)}$.
7. $S(P(U_n^*)||ID(U_n^*)||ID(BIMM_j))_{SK(BIMM_j)}$,
   $(BIMM_j \updownarrow S(P(U_n^*)||ID(U_n^*)||ID(BIMM_j))_{SK(BIMM_j)}) \rightarrow CS$.
8. $CS \leftarrow (BIMM_j \updownarrow S(P(U_n^*)||ID(U_n^*)||ID(BIMM_j))_{SK(BIMM_j)})$,
   $V(S(P(U_n^*)||ID(U_n^*)||ID(BIMM_j))_{SK(BIMM_j)})_{PK(BIMM_j)} \overset{?}{=} 1$. If it is one, then go to (9), else show "The authentication of the image of the unidentified user failed between the cloud server and the back-end image management machine", and go to (13).
9. $CS \oplus (P(U_n^*)||ID(U_n^*)||ID(BIMM_j))$.
10. $CS \triangleright \triangleleft (\exists P(U_i) \in DB(P(U_i)||ID(U_i)||ID(BIMM_j))$,
    $CS \triangleright \triangleleft (P(U_i) \overset{?}{=} P(U_n^*) \wedge ID(U_i) \overset{?}{=} ID(U_n^*))$, if $\exists P(U_i) \in (DB(P(U_i)||ID(U_i)||ID(BIMM_j)))$ and $P(U_i) = P(U_n^*) \wedge ID(U_i) = ID(U_n^*)$, then the cloud server shows "The verification is successful between the back-end image management machine and the cloud server", and go to (11), else the cloud server shows "The verification failed between the back-end image management machine and the cloud server", and go to (13).
11. $message3$ = (The verification is successful between the back-end image management machine and the cloud server),
    $(CS \updownarrow E(message3)_{PK(BIMM_j)}) \rightarrow LDMM_j, (CS \updownarrow E(message3)_{PK(BIMM_j)}) \rightarrow BIMM_j$.
12. $CS \triangleright \triangleleft$the transmission of the message is over. If it is over, then go to (13), else go to (11).
13. End.

---

The specific process execution is described as follows:

Step 1: The back-end image management machine $BIMM_j$ receives the image authentication result message sent by the cloud server and decrypts the message.

Step 2: The back-end image management machine $BIMM_j$ determines the decrypted image authentication result message, and if it is "The verification is successful between the local domain management machine and the cloud server", go to Step 3; otherwise, go to Step 14.

Step 3: The back-end image management machine $BIMM_j$ first signs the "Apply for authentication" message using the private key $SK(BIMM_j)$, i.e., ("Apply for authentication")$_{SK(BIMM_j)}$, then sends the signed message ("Apply for authentication")$_{SK(BIMM_j)}$ to the cloud server.

Step 4: After receiving the signature message ("Apply for authentication")$_{SK(BIMM_j)}$ sent by the back-end image management machine $BIMM_j$, the cloud server authenticates the signature message ("Apply for authentication")$_{SK(BIMM_j)}$ by using the public key of the back-end image management machine $BIMM_j$. If the authentication is successful, go to Step 5. Otherwise, display "Apply for authentication failure", and go to Step 13.

Step 5: The cloud server encrypts the "The authentication is successful, please transmit the image of the unidentified user" message according to the elliptic curve encryption method using the public key $PK_1(BIMM_j)$ and then sends the encrypted message ("The authentication is successful, please transmit the image of the unidentified user")$_{PK_1(BIMM_j)}$ to the back-end image management machine $BIMM_j$ via SSL or TLS.

Step 6: After receiving the encrypted message, the back-end image management machine $BIMM_j$ decrypts the message by the private key $SK_1(BIMM_j)$.

Step 7: According to the decrypted message, the back-end image management machine $BIMM_j$ first uses the private key $SK(BIMM_j)$ to sign the image $P(U_n^*)$ of the unidentified user sent by the local domain management machine. Then, sign its corresponding user identity information, which is initially compared successfully by the local domain management machine and the back-end image management machine $BIMM_j$. Finally, sign its own identity information $ID(BIMM_j)$, i.e., $((P(U_n^*)||(ID(U_n^*)||ID(BIMM_j))_{SK(BIMM_j)}$, then send the signature message $((P(U_n^*)||(ID(U_n^*)||ID(BIMM_j))_{SK(BIMM_j)}$ to the cloud server.

Step 8: After receiving the signature message $((P(U_n^*)||(ID(U_n^*)||ID(BIMM_j))_{SK(BIMM_j)}$ sent by the back-end image management machine $BIMM_j$, the cloud server authenticates the signature message by using the public key of the back-end image management machine $BIMM_j$. If the signature authentication is successful, go to Step 9, otherwise, display "The authentication of the image of the unidentified user failed between the cloud server and the back-end image management machine", and go to Step 13.

Step 9: The cloud server stores the image $P(U_n^*)$ of the unidentified user, which has been authenticated by the signature, its corresponding user identity information $ID(U_n^*)$, and identity information $ID(BIMM_j)$ of the back-end image management machine $BIMM_j$.

Step 10: The cloud server first queries the image information $P(U_i)$ of all legal users, the identity information $ID(U_i)$ of the legal user, and the identity information $ID(BIMM_j)$ of the back-end image management machine $BIMM_j$ in the cloud server image information database $DB(P(U_i)||ID(U_i)||ID(BIMM_j))$. Then, it respectively compares the image $P(U_n^*)$ of the unidentified user, which has been authenticated by the signature and the identity information $ID(U_n^*)$ of the unidentified user image with the image information $P(U_i)$ of all legal users and the identity information $ID(U_i)$ of the legal user in the cloud server image information database $DB(P(U_i)||ID(U_i)||ID(BIMM_j))$. If there is image information $P(U_i)$ of a legal user and identity information $ID(U_i)$ of a legal user in the cloud server image information database $DB(P(U_i)||ID(U_i)||ID(BIMM_j))$, this image information $P(U_i)$ and identity information $ID(U_i)$ are respectively the same as the image $P(U_n^*)$ of the unidentified user, which has been authenticated by the signature and the identity information $ID(U_n^*)$ of the unidentified user image, i.e., $(P(U_n^*) = P(U_i), ID(U_n^*) = ID(U_i))$, and the cloud server displays "The verification is successful between the back-end image management machine and the cloud server". Otherwise, the cloud server displays "The verification failed between the back-end image management machine and the cloud server", and it goes to Step 13.

Step 11: The cloud server encrypts the image authentication result message "The verification is successful between the back-end image management machine and the cloud server" and then sends the encrypted message to the local domain management machine and the corresponding back-end image management machine $BIMM_j$ via SSL or TLS.

Step 12: The cloud server determines whether the image authentication result message is sent. If sent, go to Step 13; otherwise, go to Step 11.

Step 13: End.

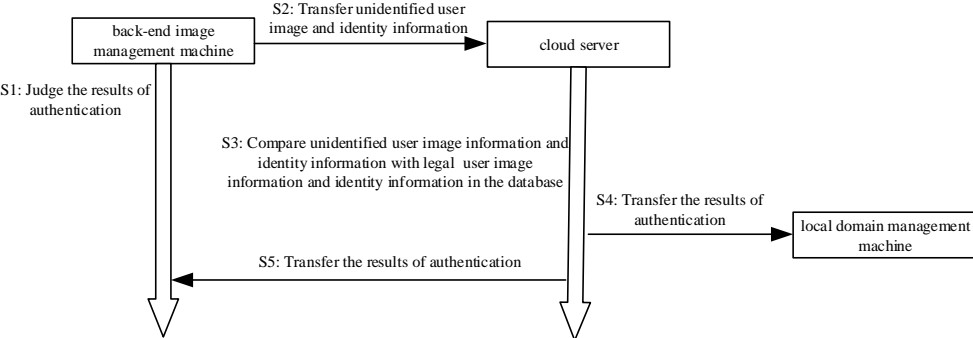

**Figure 5.** The authentication phase of the back-end image management machine and cloud server brief process diagram.

## 4. Security and Cost Analysis

In this section, firstly, we analyze the security of the multi-point collaborative authentication method based on user image intelligent collection and compare our method with Tai's method [5] and Kalra's method [15]. Secondly, we analyze the cost of our method, and the specific description is as follows. The analysis of safety functions is shown in Table 2.

### 4.1. Security Analysis

**Table 2.** Analysis of safety functions.

| Security Function | Our Method | Tai's Method | Kalra's Method |
|---|---|---|---|
| Resist replay attack | √ | √ | √ |
| Resist denial of service (DOS) attack | √ | √ | √ |
| Resist server camouflage attack | √ | × | √ |
| Resist counterfeit attack | √ | × | √ |
| Resist eavesdropping attack | √ | √ | √ |
| Resist password guessing | √ | × | × |
| Resist smart card attacks | √ | × | √ |
| Multi-point mutual authentication | √ | × | × |

4.1.1. Resist Replay Attack and Denial of Service Attack

In this method, in the authentication phase of the local domain management machine and the back-end image management machine, the front-end image collector $FIIC_k$ uses the timestamp value $T_i$ when collecting the image information of the unidentified user, and after the local domain management machine receives the image of the unidentified user sent by the front-end image intelligent collector $FIIC_k$ and the current timestamp value $(P(U_n^*)||T_i)$, another timestamp value $TS_i$ is generated. The local domain management machine first checks the freshness of the time stamp value, that is whether $T_i - TS_i$ is within the allowable time interval $\Delta t$. If $(T_i - TS_i) \geq \Delta t$, the session times out, and the image $P(U_n^*)$ of the unidentified user sent by the front-end image intelligent collector $FIIC_k$ is deleted. Assuming that the attacker replays the image information $P(U_n^*)$ and the timestamp value $T_i$ that have been verified by the local domain management machine, the local domain management machine can judge the freshness of the generated different timestamp values $TS_i$, ignoring the duplicated information, against these replay attacks. It can also reduce the consumption of network bandwidth and resist denial of service attacks.

### 4.1.2. Resist Server Camouflage Attack and Counterfeit Attack

In this method, a malicious attacker cannot masquerade as the local domain management machine to send the image $P(U_n^*)$ of the unidentified user to the back-end image management machine for authentication and cannot send fake $P(U_n^*)$ information to defraud authentication. Before sending the $P(U_n^*)$ to the back-end image management machine, the local domain management machine first signs the "Apply for authentication" with the private key $SK(LDMM_j)$ and sends it to the back-end image management machine. The back-end image management machine uses the public key $PK(LDMM_j)$ to decrypt the information. Then, it determines whether the application is sent by the local domain management machine. It can prevent the attacker from disguising the local domain management machine to destroy the authentication and resist the server camouflage attack. After the message is successfully authenticated, the back-end image management machine uses the ellipse encryption algorithm to encrypt the "The authentication is successful, please transmit the image of unidentified user" message and sends it to the local domain management machine. Finally, the local domain management machine uses the private key $SK(LDMM_j)$ to sign the $P(U_n^*)$ and sends it to the back-end image management machine. The back-end image management machine determines whether the $P(U_n^*)$ is sent by the local domain management machine by decrypting the information with the public key $PK(LDMM_j)$. It can prevent the attacker from impersonating $P(U_n^*)$ information to destroy the authentication and resist the counterfeit attack. Similarly, the authentication phase of the local domain management machine and the cloud server and the authentication phase of the back-end image management machine and the cloud server need to be authenticated in this form. It can also resist server camouflage attack and counterfeit attack. In Tai's method, the sensor node is exposed in public. If a malicious attacker destroys any node, then he/she can pretend that the user is logged into the normal legal sensor node and launch a counterfeit attack on other sensor nodes.

### 4.1.3. Resist Eavesdropping Attack and Password Guessing

This method does not use passwords for identity authentication. The only identity information is legal user image information $P(U_i)$. During the system establishment phase, the malicious attacker cannot steal the legal user image information $P(U_i)$ sent by the back-end image management machine $BIMM_j$ to the local domain management machine and the cloud server from the common channel. Because the legal user image information $P(U_i)$ is encrypted by the ellipse encryption algorithm in the common channel, the attacker cannot calculate the $P(U_i)$ information in polynomial time. In Tai's method, the user's password is stored in the smart card. Once the smart card is stolen by an authorized malicious attacker, he/she can guess and calculate the actual password of the smart card owner. In Kalra's method, a malicious attacker first guesses the password and calculates it to verify that the password is the correct one. If not, repeat the guess. The attacker can guess the correct password in a brute force way.

### 4.1.4. Resist Smart Card Attacks

In this method, image intelligent collection technology is used for identity authentication. Compared with the traditional smart card authentication method, the image intelligent collection method is less expensive and has better security and portability. There is no risk of lost, stolen, or duplicated smart cards, and there is no need to defend against attackers' attacks on smart card data. In Tai's method, user information is stored in a smart card. Once a smart card is stolen or lost, a malicious attacker can extract all the private information stored in the smart card.

### 4.1.5. Multi-Point Collaborative Authentication

This method uses a method of mutual authentication between the local domain management machine and the back-end image management machine, the local domain management machine and the cloud server, and the back-end image management machine and the cloud server. First, in the

authentication phase of the local domain management machine and the back-end image management machine, the local domain management machine needs to compare the unidentified user image $P(U_n^*)$ sent by the front-end image collector with the legal user image $P(U_i)$ in the database constructed in the system establishment phase. If the same (i.e., $P(U_n^*) = P(U_i)$), the local domain management machine encrypts the unidentified user image $P(U_n^*)$ and transmits it to the back-end image management machine. Finally, compare it with the legal user image $P(U_i)$ in the database built internally to complete the phase authentication. The principle of the authentication phase of the local management machine and the cloud server and the principle of the authentication phase of the back-end image management machine and the cloud server are similar. If a privileged attacker steals or modifies the database information in some way and destroys the authentication at a certain phase, it cannot pass the collaborative identity authentication. In Tai's method, the sensor nodes, gateway nodes, and users provided by the company cannot authenticate each other. In Kalra's method, the embedded devices and cloud servers provided by them cannot mutually confirm the legitimacy of each other. Therefore, their methods are more vulnerable to spoofing attacks.

*4.2. Cost Analysis*

The method uses the symmetric encryption algorithm and the asymmetric encryption algorithm. The characteristics of the symmetric encryption algorithm and the asymmetric encryption algorithm show that the symmetric encryption algorithm uses the same key for encryption and decryption, and the operation is fast, but easy to crack. The asymmetric encryption algorithm uses public key encryption and private key decryption, which is slow, but not easy to crack. The calculation of the operation of an asymmetric encryption algorithm ($A$) is equivalent to a point operation and is also equal to 1000 symmetric encryption algorithm operations ($S$). Therefore, assuming that the calculation cost of the asymmetric encryption algorithm operation ($A$) is one, the calculation cost of the symmetric encryption algorithm operation ($S$) is 0.001. The results of the cost analysis at different phases are shown in Table 3.

**Table 3.** Method costs. *S*, symmetric; *A*, asymmetric.

|  | *N* **Legal Identity Users in the System and the Operating Cost of Collecting** *M* **Unidentified Users** |
| --- | --- |
| System establishment phase | $4S \times N = 0.004N$ |
| The authentication phase of local domain management machine and back-end image management machine | $(6S + 1A) \times M = 1.006M$ |
| The authentication phase of local domain management machine and cloud server | $(6S + 2A) \times M = 2.006M$ |
| The authentication phase of the back-end image management machine and cloud server. | $(6S + 2A) \times M = 2.006M$ |
| Complete method | $(18S + 5A) \times M + 4S \times N = 5.018M + 0.004N$ |

## 5. Conclusions and Future Work

This paper proposed a multi-point collaborative authentication method based on user image intelligent collection in IoT. The method mainly consisted of four phases, namely the system establishment phase of user image intelligent multi-point collaborative authentication, the authentication phase of the local domain management machine and back-end image management machine, the authentication phase of local domain management machine and cloud server, and the

authentication phase of the back-end image management machine and cloud server. To demonstrate the validity of the method for identity authentication, a series of security analyses was conducted. Compared with the traditional single identity authentication method, our method used three-party collaborative authentication to avoid the problem of sensor information stealing easily caused by a single authentication method, which makes the user's identity authentication more secure and effective. The analysis results showed that the method was able to resist multiple types of attacks to meet the security requirements, attacks such as replay attacks, denial of service attacks, and server camouflage attacks. In addition, the results also indicated that the method was suitable for identity authentication in the IoT environment.

This paper did not cover the field of image acquisition and authentication, but only compared the user's image information. Therefore, it is not yet possible to estimate the impact of image acquisition and authentication on the cost, efficiency, and security of the method. In addition, image acquisition and authentication are also affected by many factors, such as ambient lighting, which result in a reduction in the recognition rate and performance. Therefore, the future work is to optimize the algorithms for image acquisition and authentication and consider adding biometrics such as fingerprint recognition to protect the identity authentication and improve the accuracy of identity authentication.

**Author Contributions:** Conceptualization, Y.L. and Y.T.; formal analysis, Y.T.; funding acquisition, Y.L.; methodology, Y.L. and Y.T.; resources, Y.L.; supervision, Y.L. and J.L.; visualization, Y.T.; writing—original draft, Y.T. and J.L.; writing—review and editing, Y.T. and J.L.

**Funding:** This research was funded by the Zhejiang Provincial Natural Science Foundation of China Grant Number Y20F020088.

**Conflicts of Interest:** The authors declare no conflict of interest.

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
