# Peer review of "Multi-Point Collaborative Authentication Method Based on User Image Intelligent Collection in the Internet of Things"

_electronics, doi:10.3390/electronics8090978_

Round 1

Reviewer 1 Report

This manuscript presents an authentication method using multiple agents within the IoT domain. 

The references in the state of the art section must show the disadvantages with respect to this manuscript. Also, the references are inserted in individuals paragraphs without a connection between them. Rephrase all the section to be easier the reading. At the end of this section, explain why your method is better than the rest.

The results must present a comparison with another state-of-the-art method.

Reviewer 2 Report

The reviewer strongly recommend this work to be reconsidered after minor revision on journal.
However, some comment should pay attention to improve the quality of paper:

1.  In abstract, the proposed algorithm is described in brief, but it is needed to explain what makes it improve the algorithm compared to conventional techniques in detail.

2. In introduction, the contribution of paper should describe and insert into paragraph with comment in detail compared to previous work.

3. From line 31 to  47, the paragraph should move into literature review.

4. In title, user image intelligent collection is not represent and clear in concept. The concept should be depicted in detail in contents of paper.

5. In chapter 3, figure 1 should describe and represent with their protocol procedure in detail.
   We can not understand the mechanism in clear.

6. In chapter 4, performance analyses should be compared with related work.
   Authors described several research in chapter 2 with many references in advanced.
   Make a table to compare the performance such as security function with related works.

7. In chapter 4.2, cost analysis could not understand the cost of symmetric and asymmetric operation.
   the concept should be defined.

Round 2

Reviewer 1 Report

The related work must be improved for an easier flow. Compare with literature.

Reviewer 2 Report

The reviewer strongly recommend this work to be reconsider as accept on journal.

Requested comments were revised and improved to improve quality of paper:

Author Response

Thank you for your review.